# Malignant Phyllodes Tumor of the Breast and Pregnancy: A Rare Case Report and Literature Review

**DOI:** 10.3390/medicina58010036

**Published:** 2021-12-26

**Authors:** Laura Mustață, Nicolae Gică, Radu Botezatu, Raluca Chirculescu, Corina Gică, Gheorghe Peltecu, Anca Maria Panaitescu

**Affiliations:** 1Obstetrics and Gynecology Department, Carol Davila University of Medicine and Pharmacy, 11171 Bucharest, Romania; lauramustata@yahoo.com (L.M.); radu.botezatu@umfcd.ro (R.B.); corina.mat@gmail.com (C.G.); gheorghe.peltecu@umfcd.ro (G.P.); anca.panaitescu@umfcd.ro (A.M.P.); 2Filantropia Clinical Hospital, 11171 Bucharest, Romania; ralucachirculescu@gmail.com

**Keywords:** phyllodes tumor, breast, borderline, malignant, pregnancy

## Abstract

Phyllodes Tumor (PT) is a rare fibroepithelial breast tumor that can behave differently depending on its biologic features. Traditionally, PTs are classified by their histologic features into benign, borderline, and malignant. In most cases that were reported, all PTs may recur, but only the borderline and malignant PT can metastasize. PT usually occurs as a breast lump or accidental finding on ultrasound (US) examination. The clinical features include a well-defined breast mass, regular or lobulated. The diagnosis is based on the integration of morphology features, but remains challenging, particularly in the distinction from fibroadenomas. We report a case of a 36-year-old patient who presented for a voluminous breast mass, rapidly growing in the past 3–4 months. At presentation, the patient was 19 weeks pregnant. The breast tumor had the clinical and US aspect of PT. A core needle biopsy was obtained, confirming a benign PT, and local excision was performed with no postoperative complications. The final pathology report showed a borderline PT with close resection margins of 1 mm. Immunohistochemistry (IHC) established the diagnosis of malignant PT with heterologous sarcomatous differentiation. The case was discussed in the multidisciplinary tumor board (MDT) and mastectomy was recommended. The patient fully consented but refused surgery at 25 weeks’ gestation, fearing premature delivery. The right breast was closely monitored by US, and at 9 weeks after the first surgery, signs of local recurrence were detected. At 35 weeks’ gestation, right mastectomy was performed, with no perioperative complications. The pregnancy was closely followed up and no complication were found. The final pathology report describes multiples PT recurrences with heterologous sarcomatous differentiation. The pregnancy outcome was uneventful, and the patient delivered a healthy child vaginally at term with no peripartum complication. Postpartum, a computer tomography (CT) examination of the head, thorax, abdomen and pelvis was performed, with no evidence of metastases. Adjuvant chemotherapy and radiotherapy completed the treatment. The follow-up and CT scan showed no metastases or further recurrence 4 years after diagnosis. In conclusion, diagnosis of PT can be difficult, especially because of the easy confusion with fibroadenoma of the breast. There are rare cases when a pathology exam needs further assessment and IHC is recommended for accurate diagnosis. Although malignant PT is rare and accounts for <1% of all breast cancers, the diagnosis and treatment that are recommended are based on the reported cases. Moreover, when complete surgical excision is achieved, the rates of recurrence and distant metastases are low, and adjuvant therapy might not be necessary.

## 1. Introduction

Phyllodes tumor (PT) is a rare breast tumor with heterogenous behavior depending on the pathology features. Benign PTs can mimic breast fibroadenoma but can recur if they are not excised with wide margins. Moreover, some PTs can present with borderline biology or can behave as malignant sarcomatous breast tumors that easily metastasize distantly [1,2]. Traditionally, PTs are classified by their histologic features into benign, borderline, and malignant. According to cases that have been reported, all PTs may recur, but only the borderline and malignant PT can metastasize [3]. PT was originally described by Johannes Muller in 1838 as “cystosarcoma phyllodes”, but nowadays the World Health Organization (WHO) has adopted the term PT, after more than 60 names that were used to describe these tumors [2,4]. The average incidence per year in the U.S.A. is around 2.1 per million women. Latina white women seems to have a higher rate of PTs compared with non-Latina white, Asian, and African American women [5,6]. PTs are associated with Li Fraumeni syndrome, a rare syndrome characterized by the development of multiple sarcomatous tumors [4,5,7,8,9].

PTs usually occur as a breast lump or accidental finding on an ultrasound (US) examination. Clinical features include a well-defined breast mass, regular or lobulated. Usually, the breast lump is firm, mobile, and painless, with a tumor size ranging from 1 cm to more than 10 cm (median size 4 to 7 cm). Other changes, like nipple retraction, ulceration, nipple discharge, or tumor fixation on chest wall are common [2,10]. However, axillary metastases are uncommon, and most of the palpable axillary lymph nodes are reactive, non-metastatic. Around 20% of PTs are diagnosed by screening mammograms as lobulated masses, very similar to fibroadenomas on imaging [2,7,11].

A diagnosis of PT should be suspected in large, rapidly growing breast tumors with fibroadenomas features on imaging. Core needle biopsy is standard for microscopic diagnosis of PT [6]. Standard imaging is recommended for breast lumps, such as mammogram and ultrasound evaluation. US imaging shows a hypoechoic, solid, well-defined mass. An integration of clinical features, such as large tumor, rapidly growing, should raise the suspicion of PT [2]. Breast MRI is usually offered for selected cases to document the extent of the disease and also, to check the resectability of the tumor [12,13].

Pathology examination remains the gold standard for the diagnosis of PT. A core biopsy or excisional biopsy are used for diagnosis. Fine needle aspiration is not recommended for PT because it is less accurate and may not be diagnostic [6]. The diagnosis of PT is based on the integration of morphology features, but remains challenging, particularly in the distinction from fibroadenomas.

Malignant PT is characterized by marked stromal cellularity and nuclear pleomorphism, stromal overgrowth, and more than 10 mitoses per 10 HPF (compared with 3 mitosis/10 HPF for benign PT). The presence of heterologous sarcomatous elements (liposarcoma, chondrosarcoma, and osteosarcoma) alone qualifies a PT as malignant. The differential diagnosis of malignant PT includes sarcomas and metaplastic (sarcomatoid) carcinoma. Immunohistochemistry may be helpful, but there are some specific microscopic feature that can distinguish between metaplastic carcinoma and sarcomas. The presence of leaf-like architecture and bland epithelium lining cleft-like spaces is typical of PT, whereas metaplastic (sarcomatoid) carcinoma is characterized by malignant epithelial elements. Immunohistochemistry elements, such as cytokeratins (CKs) (CKAE1/AE3, CK5/6, 34βE12, cam 5.2) and myoepithelial marker p63 should be used for the workup because of variable staining patterns in metaplastic (sarcomatoid) carcinomas. The majority of PTs are negative for CKs and p63 [14]. Studies show that the overall rate of correctly diagnosed fibroadenoma and PT ranges from 40% to 60%. Molecular features such as p53, ki67, EGFR, and PDGF have failed to predict outcomes. Moreover, receptors for estrogen and progesterone are commonly used for epithelial tumors, and are not useful for stromal component in PT [10].

Standard treatment for PT includes complete surgical excision. No axillary dissection is recommended [14,15]. Radiotherapy and chemotherapy remain as options for adjuvant treatment in selected cases, only for borderline or malignant tumors. Moreover, chemotherapy is selected for recurrent, high-risk patients. Hormonotherapy is not used in PT tumors. Taking into consideration the rarity of the disease, the guidelines used for PT are the results of cases reported so far retrospectively [6,14]. Surgery is the first line treatment for PT—complete surgical excision being recommended because some studies have shown that positive margins are associated with higher local/distance recurrence rates (fourfold higher for positive margins). A positive margin needs further re-excision. In some retrospective studies in women with malignant PT, mastectomy and breast-conserving surgery were performed almost equally (48% vs. 52%), wide local excision being associated with the same or improved survival rate, regardless of the tumor size [15]. Adjuvant radiotherapy is recommended for cases where clear margins were not achieved, especially for borderline and malignant tumors [16,17]. The role of systemic chemotherapy is limited. There is still controversy in use of chemotherapy for malignant PT. Although adjuvant chemotherapy is rarely recommended, we should consider it for high-risk patients. Criteria to establish high-risk patients were considered to be tumors > 10 cm or recurrent/metastatic disease, but this is a field to explore further, since other classification might need to be added for a more comprehensive approach. Therefore, additional stratification is expected to come from future studies. Usually, the chemotherapy regimens used for malignant PT are similar to soft-tissue sarcoma guidelines [18].

The follow-up guidelines are not standard given the rarity of the disease. Therefore, it is recommended to use the follow-up guidelines from soft-tissue sarcoma for close surveillance and efficient rate of detection of recurrence or metastases.

## 2. Case Presentation

We report a case of a 36-year-old patient who presented to the hospital for a rapidly growing tumor in the right breast that had developed and enlarged considerably in the previous 3–4 months. At presentation, the patient was 19 weeks pregnant. Her medical history included an uncomplicated vaginal birth and a surgical excision of a right breast fibroadenoma (18 years prior) and she had no significant family history of breast cancer or other types of cancer.

On clinical examination, a large tumor of 10/8 cm was found in the upper-outer quadrant of the right breast. It was painless, mobile, and lobulated, with no skin changes. The US examination described a large, non-homogenous nodule in the central and upper-outer quadrant of the right breast, with a diameter of 10 cm and small cystic areas. There was no evidence of axillary lymph nodes on clinical examination and US evaluation. The US examination raised suspicion of PT and a core needle biopsy was performed. The pathology report confirmed a benign PT. Surgical treatment was recommended, taking into consideration the size of the tumor and the rapidly growing pattern. Breast-conserving therapy was performed with excision of the breast tumor under general anesthesia with very good cosmetic result and no postoperative complications (Figure 1). The final pathology examination showed borderline PT (Figure 2), with 1 mm excision margin (Figure 3). Further immunohistochemistry (IHC) confirmed malignant PT (with heterologous sarcomatous differentiation) (Figure 4).

The case was discussed in the multidisciplinary tumor board (MDT), and taking into consideration the size of the tumor, the breast/tumor ratio, and the constitutional factors, breast conservation could not achieve clear margins. Mastectomy was recommended, and the patient was fully informed. She refused surgery at 25 weeks’ gestation because she did not want to risk premature delivery. The patient was closely monitored by US examination every 4 weeks. Nine weeks after the first surgery, US exam showed local recurrence and progression of the disease. The US described at least 6 parenchymatous right breast nodules, the largest one with a diameter of 2.8 cm (Figure 5). No other abnormalities were found in the contralateral breast or in the axillae. At 35 weeks gestation, right mastectomy was performed, with no perioperative complications. Pregnancy was closely followed up, without other complications. The final pathology report described multiples tumors (Figure 6) with heterologous sarcomatous differentiation (Figure 7), the largest one with a diameter of 4/2 cm.

The patient recovered well, and 4 weeks later she delivered a healthy child, 3600 g. A postpartum computer tomography examination of the head, thorax, abdomen, and pelvis was performed, with no evidence of metastases. The patient received adjuvant chemotherapy, consisting of 6 series of Doxorubicin, and radiotherapy. Four years after diagnosis, the patient has no sign of local recurrence or distant metastases. Three years after mastectomy, the patient underwent a right breast reconstruction with good cosmetic result. Close follow-up is done regularly.

## 3. Discussion and Conclusions

Diagnosis of PT can be difficult, especially because of the easy confusion with fibroadenoma of the breast. The tumors look similarly on imaging and clinical examination. The histology can bring more information to differentiate them; however, there are rare cases when a pathology exam needs further assessment, and IHC is necessary for accurate diagnosis. Although malignant PT is rare and accounts for <1% of all breast cancers, the recommended diagnosis and treatment are based on reported cases. Moreover, when complete surgical excision is achieved, the rates of recurrence and distant metastases are low, and adjuvant therapy might not be necessary.

After pathology examination, the standard treatment recommended by guidelines depends on the grade of the tumors; however, surgical excision with clear margins remains the gold standard for PT cases. For malignant and borderline tumors, there is a higher risk of recurrence and metastasis when less than 1 cm margins are achieved—therefore re-excision is recommended when margins are not clear.

The association of pregnancy with malignant PT is very rare. Management should be individualized. Prognosis data shows that approximately 10% to 15% of PTs are malignant, and the local recurrence rate ranges from 15% to 40%. Studies have shown that 9% to 27% of malignant PTs metastasize to distal organs. Most patients with metastasis do not respond to standard chemotherapy and die within 3 years of initial treatment. Phyllodes tumors most often metastasize to the lungs, tumors >5 cm having a poor prognosis [14]. Recent studies have shown that malignant Phyllodes can promptly respond to chemotherapy in the vast majority of eligible cases; however, the responses are not substantial and the risk of recurrence and metastasis is persistent [19,20].

Although there is no specific guideline for the treatment of PT in pregnancy, and in most cases, treatment is guided by reports from the literature, there are a few cases mentioned in the literature that showed surgery alone might not be enough for selected cases regarding rate of recurrence and metastasis. Surgery (excision with >1 cm margin) remains the gold standard and the main recommendation for these cases, but when this is not achievable, chemotherapy and radiotherapy seems to lower the rate of recurrence and prevent progression of the disease. However, oncological treatment should take into consideration the pregnancy, the timeline regarding the delivery of the baby, and both patients, mother and baby, equally. It is well known that chemotherapy is a good option, when needed, for second-trimester pregnancy. On the other hand, radiotherapy is done postpartum and, therefore, some specific cases might need to take into consideration preterm delivery in order to complete the adjuvant treatment in a recommended timeline for preventing recurrence. Surgery and general anesthesia are also considered safe during pregnancy when the benefits exceed the risks for the future mother.

In the literature, we found less than 10 cases of singular case reports of malignant PT in pregnancy and lactation, and a systematic review that included 37 studies with 43 cases [21]. The latter included all types of PT in pregnancy, 60% (26 cases) of these being malignant. From the cited malignant cases, some were not evaluated by US, and the diagnosis was made after pathology examination on the surgical specimen. However, from the cases that were described by US imaging, there were no specific findings to discriminate a benign PT from a malignant tumor, since contour irregularities were seen in both types of tumors, and positive Doppler scans were equally distributed between benign and malignant PT. Therefore, the diagnosis was made retrospectively on the surgical specimen [21]. Although mastectomy is likely to be recommended for some locally advanced tumors, breast-conserving surgery remains a good option for cases were surgical clear margins can be achieved. The vast majority of cases described conservative surgery as a first approach, but for most of the malignant cases, mastectomy was recommended in order to achieve clear margins as completion surgery [22,23,24,25]. Fatal outcome is uncommon for these diagnoses, since most of the tumors have a rapid growth pattern during pregnancy, and usually patients will approach a doctor sooner for diagnosis. However, there is a case reported of a fatal outcome of malignant PT diagnosed during pregnancy. The authors highlight the importance of early diagnosis and treatment. For this case particularly, the progression of the disease was shown soon after the first surgical excision, and the lung metastasis was fatal one year after the first operation, taking into consideration that the excision was performed 2 years after the patient first felt the lump [26].

A systematic review described 26 cases of malignant PT during pregnancy and lactation (3 cases of bilateral malignant PT), from which 19 cases underwent mastectomy, but 8 of those were dealt with a conservative approach first, and mastectomy was performed after microscopic positive margins, or when recurrence occurred. Only one patient had breast reconstruction. Pregnancy outcome was not affected by the diagnosis, and all patients delivered healthy babies at term. One patient was treated with adjuvant chemotherapy and radiotherapy and another patient received adjuvant radiotherapy alone. Patients were followed up at least 12 months (for 13 out of 26 patients) after diagnosis and the prognosis was generally good. Six of these cases had local recurrence or distant metastases, and two patients died 2 years and 11 years after diagnosis from recurrence or widespread metastases [21,27,28]. Although exposure to female sex hormones might incline towards malignant transformation of PT, only 2 out of 26 patients have had hormonal stimulation for infertility, so this hypothesis is difficult to be sustain, since data is limited for these cases. Further data is needed, since there has been no comparison made between malignant PT for pregnant and non-pregnant patients who undergo infertility stimulation treatment [21,28].

Malignant PT during pregnancy remains a rare entity, with good prognosis for early- diagnosed cases. However, a multidisciplinary approach is needed for these cases, taking into consideration the risk of recurrence and metastases for large tumors when clear margins are not easy to achieve. Moreover, pregnancy might hide this diagnosis, since lactation changes can conceal breast lumps, or they may become evident only when the dimensions have become extreme large. Surgery remains the gold-standard approach, and adjuvant oncological treatment is reserved for specific cases with higher risk of recurrence and metastases.

In conclusion, malignant PT should be suspected for patients with rapidly growing breasts tumors, especially during pregnancy, and early diagnosis and treatment can significantly change the prognosis of these cases.

## Figures and Tables

**Figure 1 medicina-58-00036-f001:**
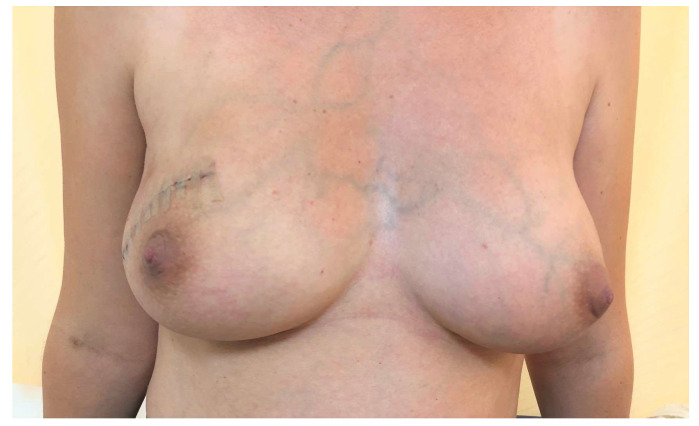
Early postoperative aspect after wide local excision of the tumor of the right breast.

**Figure 2 medicina-58-00036-f002:**
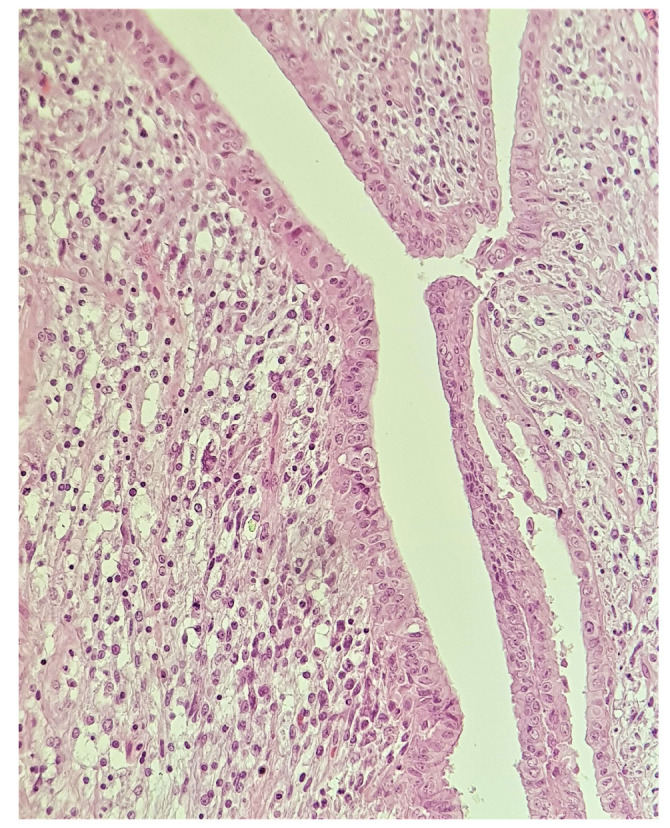
Slit-like space lined by benign epithelium with periductal condensation of atypical stromal cells (H&E stain, ob. ×200).

**Figure 3 medicina-58-00036-f003:**
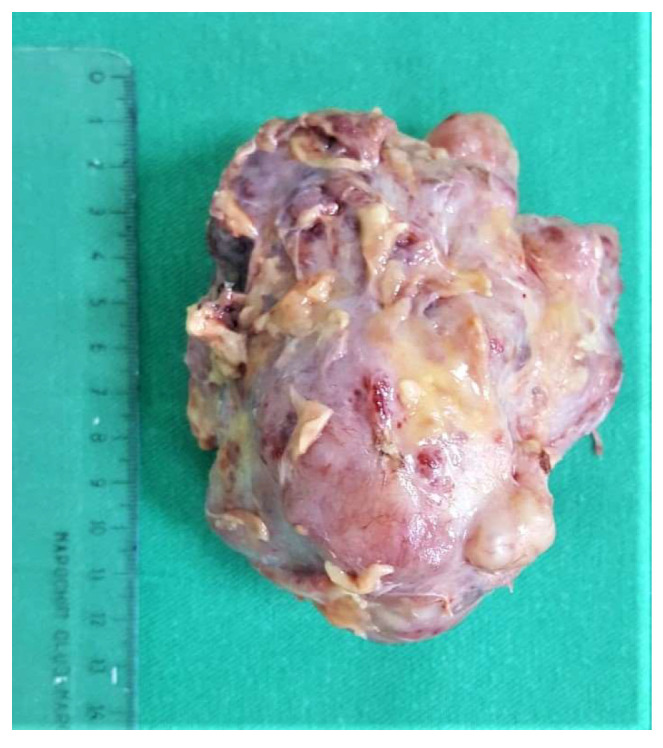
Macroscopic aspect of the right breast tumor.

**Figure 4 medicina-58-00036-f004:**
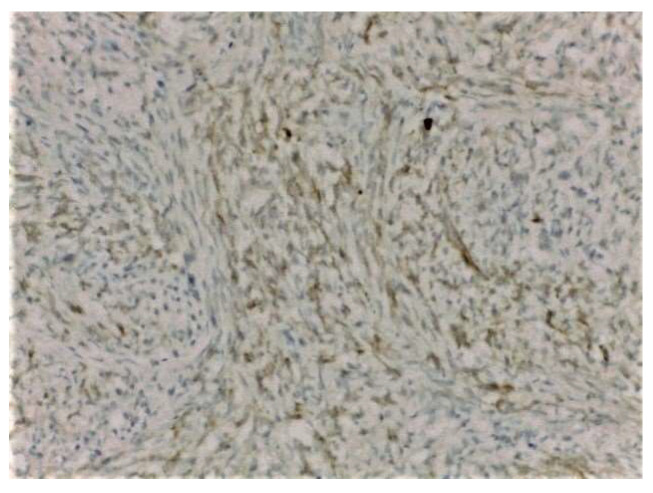
Positive expression of CD 117 in malignant Phyllodes tumor (IHC, ×200).

**Figure 5 medicina-58-00036-f005:**
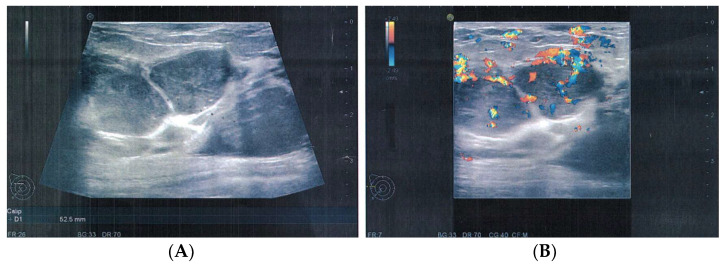
Ultrasonography aspect of local recurrence. (**A**). lobulated, well-circumscribed masses, with smooth margins. (**B**). hypervascularity on color Doppler.

**Figure 6 medicina-58-00036-f006:**
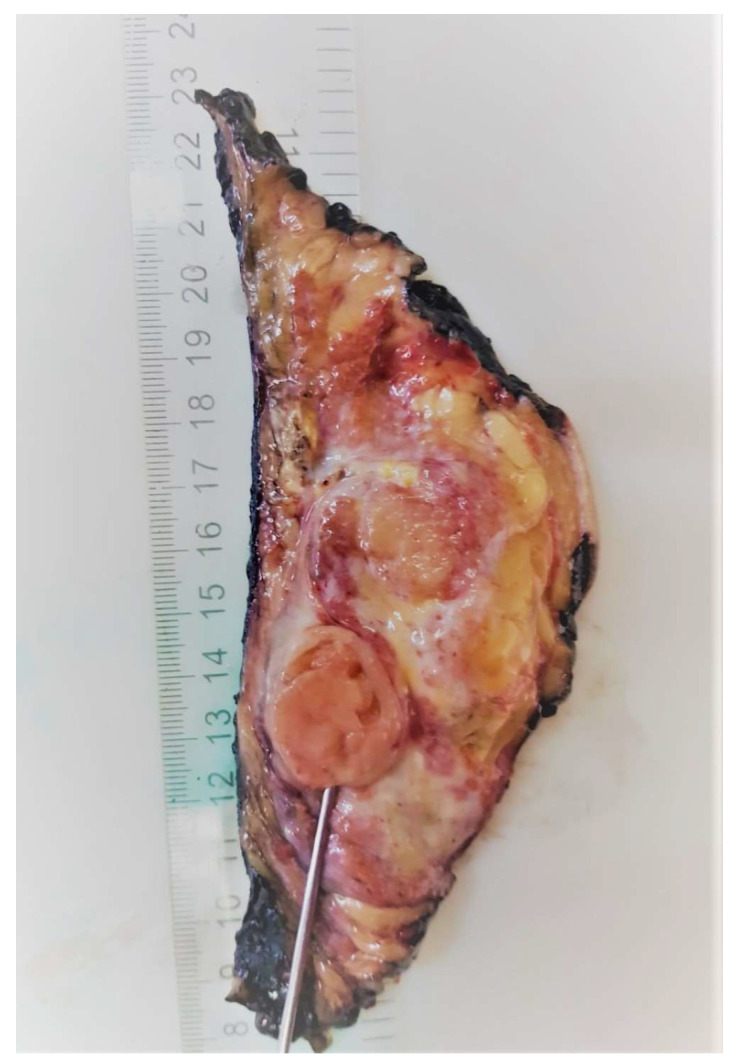
Mastectomy specimen with macroscopic aspect of the local recurrences.

**Figure 7 medicina-58-00036-f007:**
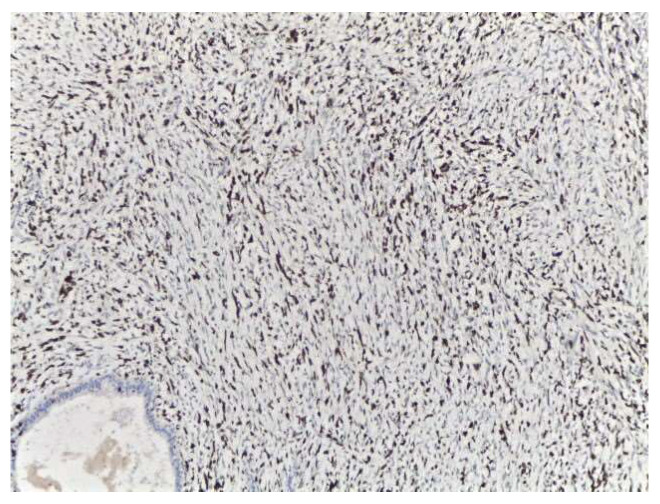
Ki67 immunostaining in malignant Phyllodes tumor (×100).

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
