# Peer review of "Malignant Phyllodes Tumor of the Breast and Pregnancy: A Rare Case Report and Literature Review"

_medicina, 2021, doi:10.3390/medicina58010036_

Round 1

Reviewer 1 Report

Page 3 Line 18: More accurate to state that criteria for who should recieve adjuvant therapy are poorly defined (>10 cm is not a commonly acknowledged cutoff, but not unreasonable)

Page 6 line 196: Tumors are often responsive to chemotherapy, but these responses are not durable.  See:  Parkes et al. 2021 Breast Cancer Research and Treatment, Mitus et al. 2016 World J Surg

Page 6 218: Mastectomy is likely recommended for locally advanced tumors in the reported cases.  But standard or care for phyllodes tumors, or even other breast sarcomas, does not preclude breast conservation therapy.

Given that this is a case of malignant PT arising in pregancy, a brief discussion of risks or safety of surgery/radiation/chemotherapy prior to delivery may be of interest.

The case report is clearly written.  Literature review is adequate but does not provide significant additional synthesis.    

Author Response

  • Page 3 Line 18: More accurate to state that criteria for who should recieve adjuvant therapy are poorly defined (>10 cm is not a commonly acknowledged cutoff, but not unreasonable)

Thank you for your kind recommendation. The changes have been made according to your suggestions as follows:

"Although adjuvant chemotherapy is rarely recommended, we should consider it for high-risk patients. Criteria to establish high risk patients were considered to be tumors >10 cm or recurrent/metastatic disease, but this is a field to explore further, since other classification might need to be added for more comprehensive approach. Therefore, additional stratification is expected to come from future studies. Usually the chemotherapy regimens used for malignant PT are similar to soft tissue sarcomas guidelines [18]. "

  • Page 6 line 196: Tumors are often responsive to chemotherapy, but these responses are not durable.  See:  Parkes et al. 2021 Breast Cancer Research and Treatment, Mitus et al. 2016 World J Surg

We have also added the data from the newer studies and the correct citations

"Recent studies showed that malignant Phyllodes can promptly respond to chemotherapy in the vast majority of eligible cases, however the responses are not substantial and the risk of recurrence and metastasis is persistent. [19,20]"

  • Page 6 218: Mastectomy is likely recommended for locally advanced tumors in the reported cases.  But standard or care for phyllodes tumors, or even other breast sarcomas, does not preclude breast conservation therapy.

We have emphasized the importance of surgical options, both mastectomy and breast conserving therapy for malignant Phyllodes tumors depending on clinical stratification of patients "Although mastectomy is likely to be recommended for some locally advanced tumors, breast conserving surgery remains a good option for cases were surgical clear margins can be achieved. The vast majority of cases described conservative surgery as first approach, but for most of the malignant cases mastectomy was recommended in order to achieve clear margins as completion surgery "

  • Given that this is a case of malignant PT arising in pregancy, a brief discussion of risks or safety of surgery/radiation/chemotherapy prior to delivery may be of interest.

The case report is clearly written.  Literature review is adequate but does not provide significant additional synthesis.   

We have also added discussion regarding safety of oncological and surgical treatment during pregnancy for these particular cases:

"However, oncological treatment should take into consideration the pregnancy, the timeline regarding the delivery of the baby and both patients, mother and baby equally. It is well known that chemotherapy is a good option when needed, for second trimester pregnancy. On the other hand, radiotherapy is done postpartum and therefor, some specific cases might need to take into consideration preterm delivery in order to complete the adjuvant treatment in a recommended timeline for preventing the recurrence. Surgery and general anesthesia are also considered safe during pregnancy when the benefits exceed the risks for the future mother."

Reviewer 2 Report

I enjoyed reading the manuscript and thank the authors for their valuable report and discussion. The paper has some minor grammatical errors or typos that should be corrected (e.g. line 162 no complication were found, line 163 multiples PT, ...).

Author Response

I enjoyed reading the manuscript and thank the authors for their valuable report and discussion.

Thank you for your comments and appreciations.

The paper has some minor grammatical errors or typos that should be corrected (e.g. line 162 no complication were found, line 163 multiples PT, ...).

Thank you for your suggestion. We modified the article accordingly.

Reviewer 3 Report

-what about elastography?  could have helped ? 

-could be considered preoperative MRI for surgical management? 

  • postoperative monitoring by which methods are recommended given the aggressive nature of the tumor?
  • what does the literature say about preoperative and postoperative imaging methods about this pathology even in non-pregnant patients what are the contraindications for preoperative MRI examination in this case?

Author Response

Thank you for your kind suggestions. We appreciate your effort during the process of publication of our paper.

  1. Although elastography is not a standardized method used for diagnosis of Phyllodes tumor, we appreciate that it might be useful for selected cases to discriminate between fibroadenoma or Phyllodes tumor of breast. (This method is still in debate for cost efficiency and at the time of diagnosis we did not have the opportunity to take into consideration elastography for this patient).

  1. MRI was taken into consideration for this particular patient, but given the pregnancy, the patient refused further imaging and declared to be claustrophobic.

  1. Postoperative management and monitoring was done by ultrasound during pregnancy and after the completion surgery and delivery she underwent under surveillance with the oncology team having scans for local and distant recurrence, by ultrasound, MRI and CT scan.

  1. When Phyllodes tumor is suspected MRI should be added after US examination, taking into consideration that most patients are under 40 at diagnosis, and mammogram would not add important information for determination the grade of suspicion. MRI can help to determine the type of Phyllodes tumor, but unless biopsy or excision is done, and taking into consideration that most of the cases were studied retrospectively, it is difficult to establish specific guidelines for these cases. However, for this particular case MRI was not possible since the patient refused the investigation.

Reviewer 4 Report

Dear author,  I recommend you to shorten the article, it is too long and in my opinion sometimes reiterative.

Author Response

Dear author,  I recommend you to shorten the article, it is too long and in my opinion sometimes reiterative.

Thank you for your comment. We agree with you and we removed some repetitive information.